# Efficient Physical Mixing of Small Amounts of Nanosilica Dispersion and Waterborne Polyurethane by Using Mild Stirring Conditions

**DOI:** 10.3390/polym14235136

**Published:** 2022-11-25

**Authors:** María Echarri-Giacchi, José Miguel Martín-Martínez

**Affiliations:** Adhesion and Adhesives Laboratory, University of Alicante, 03080 Alicante, Spain

**Keywords:** waterborne polyurethane, nanosilica dispersion, physical mixing, mild mechanical stirring, phase separation, adhesion

## Abstract

Good dispersion of nanosilica particles in waterborne polyurethane was obtained by mild mechanical stirring when 0.1–0.5 wt.% nanosilica in aqueous dispersion was added. The addition of small amounts of nanosilica produced more negative Z-potential values, increased the surface tension and decreased the Brookfield viscosity, as well as the extent of shear thinning of the waterborne polyurethane. Depending on the amount of nanosilica, the particle-size distributions of the waterborne polyurethanes changed differently and the addition of only 0.1 wt.% nanosilica noticeably increased the percentage of the particles of 298 nm in diameter. The DSC curves showed two melting peaks at 46 °C and 52 °C, as well as an increase in the melting enthalpy. In addition, when nanosilica was added, the crystallization peak of the waterborne polyurethane was displaced to a higher temperature and showed higher enthalpy. Furthermore, the addition of 0.1–0.5 wt.% nanosilica displaced the temperature of decomposition of the soft domains to higher temperatures due to the intercalation of the particles among the soft segments; this led to a change in the degree of phase separation of the waterborne polyurethanes. As a consequence, improved thermal stability and viscoelastic and mechanical properties of the waterborne polyurethanes were obtained. However, the addition of small amounts of nanosilica was detrimental for the wettability and adhesion of the waterborne polyurethanes due to the existence of acrylic moieties on the nanosilica particles, which seemed to migrate to the interface once the polyurethane was cross-linked. In fact, the final T-peel strength values of the joints made with the waterborne polyurethanes containing nanosilica were significantly lower than the one obtained with the waterborne polyurethane without nanosilica; the higher the nanosilica content, the lower the final adhesion. The better the nanosilica dispersion in the waterborne polyurethane+nanosilica, the higher the final T-peel strength value.

## 1. Introduction

Due to health and safety matters, the waterborne polyurethane adhesives (PUDs) are environmentally friendly alternatives to solvent-borne polyurethane adhesives in the bonding of different materials in the textile, footwear, automotive, furniture and outdoor industries [1]. PUDs are made of spherical polyurethane particles of nanometer size dispersed in water. Upon water removal, a cross-linking of the polyurethane particles is produced, leading to a phase separation of the soft domains (constituted by the flexible polyol chains) and the hard domains (constituted by rigid urea and urethane groups) [2,3,4,5].

In some applications, a significant adhesion of the PUD is needed immediately after bond formation. The use of PUDs is limited because the complete water removal is needed for producing sufficient adhesive strength. For increasing the immediate adhesion of the PUDs, different strategies have been proposed in the existing literature, the most common being the addition of fillers [6,7]. The inorganic fillers are the most promising for improving the properties of the polyurethanes due to the good balance provided by the organic polyurethane chains (flexibility and low density) and the inorganic filler (thermal stability and mechanical strength). Several inorganic fillers including carbon blacks [8], clays [9], graphene-based materials [10] and fumed silicas [11,12,13], mainly, have been successfully added to solvent-borne polyurethanes for improving their rheological, mechanical, thermal and immediate adhesion properties. The improved properties of the polyurethane–silica materials have been ascribed to the formation of hydrogen bonds between the urethane/urea groups of the polyurethane chains and the silanol groups on the silica surface [12]. On the other hand, the incorporation of nanosilica to the polyurethane provides films with increased indentation hardness, high resistance to whitening and high permeability to water vapor [14]. Furthermore, the addition of nanosilicas has been shown to be efficient in the improvement of water and solvent resistance, and in the increase in the surface hydrophobicity of polyurethane films [15].

The use of monodispersed colloidal silica instead of nanosilica powder allows for controlling the size distribution of the inorganic component in the hybrid polyurethane–silica materials [15,16]. Furthermore, the addition of nanosilica powder to PUD is not evident because of their inefficient dispersion due to the different polarities of the polyurethane (hydrophobic) and the silica (hydrophilic) particles; this leads to the formation of silica agglomerates [17].

Different methods have been proposed for obtaining waterborne polyurethanes with well-dispersed silica particles. These can be classified into two main groups, i.e., the incorporation of the silica precursor during the synthesis of the PUD and the physical mixing of the nanosilica dispersion and the PUD. Several variations have been proposed for adding the silica precursor during the synthesis of the PUD including the sol-gel [18,19,20,21,22,23] and the in situ polymerization methods [24,25,26,27,28,29], among others. Although effective, the incorporation of the silica precursor during the synthesis of the PUD is not simple and it is difficult to scale-up at the industrial level. In this sense, the physical mixing of the nanosilica dispersion and the PUD seems facile and more industrially attractive.

There are some previous studies dealing with the addition of different amounts of distinct nanosilica aqueous dispersions to PUDs by physical mixing [30,31,32,33,34,35]. Yang et al. [30] added 20 and 50 wt.% colloidal nanosilica to a polyester-based PUD by mechanical stirring and they reported the agglomeration of the nanosilica particles caused by the interactions between their silanol groups. However, improved thermal and mechanical properties of the PUD+nanosilica materials were found. Špírková et al. [31] mixed a PUD with up to 50 wt.% aqueous nanosilica of 30 nm particle size by mechanical stirring, followed by treatment in an ultrasonic bath. Although the PUD properties were enhanced, the mixing method was not efficient for dispersing the nanosilica particles and micrometer-size silica aggregates appeared. In addition, Serkis et al. [32] prepared PUD–silica nanocomposites containing 5, 32 and 50 wt.% of two commercial nanosilicas differing in shape and particle size. They found different properties depending on the amount of nanosilica added, i.e., flexible elastic (for 5 wt.% nanosilica) to ceramic-like (for 50 wt.% nanosilica) materials. All those studies have shown that the addition of aqueous silica dispersions improved some properties of the PUDs, but always silica aggregates of micrometric size were obtained when physical mixing was used.

One way to reduce the extent of agglomeration of the silica particles in the PUD by physical mixing is through the addition of small amounts of nanosilica. Some few studies have considered the addition of small amounts of nanosilica dispersion to PUD by physical mixing. Thus, Cackić et al. [33] added 0.5–1 wt.% hydrophilic colloidal nanosilica to PUDs with different OH functionalities by mechanical stirring. Improved thermal properties were obtained by increasing the nanosilica content. In a later study, Khaokong et al. [34] added small amounts (0.5–5 wt.%) of modified poly(diallyldimethyl ammonium chloride) silica extracted from rice husk ash to PUDs, and homogeneous PUD emulsions containing 0.5–1.5 wt.% silica were obtained. However, the sedimentation of the silica was produced, and this caused a deterioration of the tensile properties.

The adhesion properties of the PUDs containing different amounts of distinct nanosilicas have been scarcely studied. All PUD+nanosilica materials were prepared by adding the nanosilica during the synthesis of the PUD. Guo et al. [35] synthesized hybrid UV-curable PUD/nanosilica materials containing up to 17.5 wt.% functionalized nanosilica, and the adhesion was evaluated by peel tests. They found that the adhesion properties of the PUD+nanosilica materials deteriorated by increasing the silica content. Heqing et al. [36] studied the adhesion properties of nanosilica-fluorinated PUD nanocomposites containing up to 2 wt.% nanosilica prepared by seed-emulsion polymerization. The T-peel strength of the nanocomposites was improved, especially when 1 wt.% nanosilica was added. In another study, Jia-hu et al. [29] synthesized nanosilica-modified PUD adhesives containing 1–3 wt.% nanosilica via in situ polymerization, and the adhesion properties were improved when the adhesive contained less than 2 wt.% nanosilica.

The analysis of the existing literature has shown that the main drawbacks of the addition of nanosilica dispersion to the PUD by physical mixing are the agglomeration of the silica particles and the limited improvement of the properties, including the adhesion properties. In our previous study [37], a nanosilica dispersion containing 1 wt.% nanosilica was added to a PUD by using three different physical mixing procedures differing in the stirring rate and the flow regime (tangential, laminar and radial). It was found that the PUD+nanosilica material made with a higher shear rate and laminar flow regime showed high homogeneous dispersion of the nanosilica particles and a greater extent of intercalation between the soft segments of the polyurethane. This led to enhanced thermal stability and moderate changes in the mechanical properties. However, unexpectedly, the better dispersion of the nanosilica in the polyurethane matrix decreased both the wettability and the final T-peel strength of the joints made with PUD+nanosilica dispersions. In other words, there was not a relationship between the degree of nanosilica dispersion and the adhesion properties of the PUD+nanosilica dispersions.

In this study, a different approach than the ones on the existing literature for preparing PUD+nanosilica dispersions with good dispersion of the nanosilica particles is proposed. Our approach consists of the addition of very small amounts (0.1–3 wt.%) of nanosilica to the PUD by physical mixing under a low stirring rate and a tangential flow regime to obtain a high dispersion of the nanosilica particles among the polyurethane chains.

## 2. Materials and Methods

### 2.1. Materials

Anionic waterborne polyurethane dispersion Dispercoll^®^ U56 (Covestro, Leverkusen, Germany) and aqueous nanosilica dispersion LUDOX^®^ AM (Grace, Columbia, ML, USA) with a nominal primary particle size of 12 nm were used.

### 2.2. Physical Mixing of the Nanosilica Dispersion and the Waterborne Polyurethane Dispersion (PUD)

Different amounts of the nanosilica dispersion containing 0.1–3 wt.% nanosilica were added to the waterborne polyurethane dispersion by physical mixing. An amount of 182.5 mL PUD was added into a 1 L baker and stirred with a Heidolph RZR 2020 stirrer (Heidolph, Schwabach, Germany) provided with an anchor type stirrer at 200 rpm for 15 min—Figure 1. Then, 0.29–8.70 mL nanosilica dispersion (equivalent to 0.1–3 wt.% nanosilica) was added drop by drop.

The nomenclature of the waterborne polyurethane dispersions containing nanosilica consists in the capital letters “PUD” followed by “+” and the letters “SiH”; the amount of nanosilica was given at the end between brackets. For example, PUD+SiH(0.5) corresponds to the waterborne polyurethane dispersion containing 0.5 wt.% nanosilica. The nomenclature of the solid waterborne polyurethanes containing nanosilica was similar, but the capital letters “PUD” were changed to “PU”. Thus, PU+SiH(0.5) corresponds to the solid waterborne polyurethane containing 0.5 wt.% nanosilica.

### 2.3. Experimental Techniques

#### 2.3.1. Characterization of the Nanosilica Powder

Nanosilica powder was obtained by drying the aqueous nanosilica dispersion at room temperature for 3 days. The resulting solid nanosilica was crushed and sieved. 

The chemical composition of the nanosilica was analyzed by infrared spectroscopy in an Alpha FT-IR spectrometer (Bruker Optik GmbH, Ettlinger, Germany). A germanium prism was used. Sixty-four scans with a resolution of 4 cm^−1^ were recorded and averaged.

The chemical composition of the nanosilica powder surface was determined by X-ray photoelectron spectroscopy (XPS) in a K-Alpha Thermo-Scientific spectrometer (Waltham, Massachusetts, USA). An Al-Kα X-ray source (1486.6 eV) was used. A spot size of 300 μm and a pass energy of 150 eV were used. The binding energies of the photopeaks were determined by setting the C1s photopeak to C-C and C-H at 285.0 eV.

The morphology of the nanosilica powder was analyzed by transmission electron microscopy (TEM) in a Jeol JEM-1400 Plus microscope (Jeol, Tokyo, Japan) by using an acceleration voltage of 120 kV.

The crystallinity of the nanosilica powder was analyzed by wide-angle X-ray diffraction in a Bruker D8-Advance diffractometer (Bruker, Etlinger, Germany) provided with a nickel filter and a Göebel mirror.

#### 2.3.2. Characterization of the Aqueous Nanosilica Dispersion and the Waterborne Polyurethane Dispersions (PUDs) without and with Different Amounts of Nanosilica

The solids content of the aqueous nanosilica dispersion and the PUDs was determined in a DBS 60-3 thermo balance (Kern & Sohn GmbH, Balingen, Germany) by heating 0.5 g dispersion at 105 °C for 15 min, followed by additional heating at 120 °C until a constant mass was obtained. For each sample, three replicates were measured and averaged.

The pH values of the aqueous nanosilica dispersion and the PUDs were measured in a pH-meter PC-501 (XS Instruments, Carpi, Italy) equipped with an XC-PC510 electrode. For each sample, three replicates were measured and averaged.

The viscosities of the aqueous nanosilica dispersion and the PUDs were measured at 22 °C in a Brookfield RD DV-I Prime viscometer (Brookfield Engineering Laboratories Inc., Stoughton, DE, USA) according to ASTM D3236-88 standard. For each sample, three replicates were measured and averaged.

The surface tensions of the aqueous nanosilica dispersion and the PUDs were measured at 22 °C in a Phywe equipment (Göttingen, Germany) by using a metallic ring 19.5 mm in diameter. The DuNouy ring method was used. For each sample, three replicates were measured and averaged.

The particle size distributions of the aqueous nanosilica dispersion and the PUDs were determined by dynamic light scattering in a Microtrac Sync equipment (Verder Scientific group, Haan, Duesseldorf, Germany).

The Z-potential values of the aqueous nanosilica dispersion and the PUDs were measured in a Nanotrac Flex (Microtrac Mrb, Verder Scientific group, Dusseldorf, Germany) combined with Stabino equipment (Colloid Metrix, GmbH, Meerbusch, Germany).

#### 2.3.3. Characterization of the Solid Waterborne Polyurethanes (PUs) without and with Different Amounts of Nanosilica

Solid polyurethane (PU) and PU+nanosilica films were obtained by placing 20 mL dispersion in square Teflon mold with the dimensions 120 mm × 200 mm × 1 mm. The water was removed at room temperature for 1 week.

The attenuated total-reflectance Fourier transform infrared (ATR-IR) spectra of the PUs were obtained in an Alpha FT-IR spectrometer (Bruker Optik GmbH, Ettlinger, Germany) by using a germanium prism. Two replicates for each sample were obtained. All ATR-IR spectra were normalized to the C-O-C stretching band at 1170 cm^−1^.

The differential scanning calorimetry (DSC) curves of the PUs were obtained in TA DSC Q100 V6.2. equipment (TA Instruments, New Castle, DE, USA). Aluminum pans containing 10–15 mg samples were heated from −80 °C to 200 °C under nitrogen atmosphere (flow rate: 50 mL/min). The heating rate was 10 °C/min. Then, a cooling run from 200 °C to −80 °C was carried out by using a cooling rate of 10 °C/min, and, finally, a second DSC heating run from −80 °C to 250 °C was carried out by using a heating rate of 10 °C/min. Two replicates for each sample were obtained.

The thermal properties of the PUs were assessed by thermal gravimetric analysis (TGA) in TGA Q500 equipment (TA Instruments, New Castle, DE, USA). An amount of 10–15 mg PU was placed in a platinum crucible and heated under nitrogen (flow rate: 100 mL/min), from room temperature up to 600 °C, by using a heating rate of 10 °C/min. Two replicates for each sample were obtained.

The rheological and viscoelastic properties of the PUs were measured in a DHR-2 rheometer (TA Instruments, New Castle, DE, USA) by temperature sweep experiments. Parallel-plates (upper plate diameter = 25 mm) geometry was used. The gap was set to 400 µm. The measurements were carried out by decreasing the temperature from 140 °C to 10 °C in a Peltier system. A cooling rate of 5 °C/min and a frequency of 1 Hz were used. The rheological experiments were performed in the region of linear viscoelasticity. Two replicates for each sample were obtained.

Thin PU films, with the dimensions 12 mm × 24 mm × 200 µm, were prepared and their mechanical properties were assessed by stress–strain tests according to ISO 37 standard. Dog-bone test specimens were cut, and the stress–strain tests were carried out in a Zwick/Roell Z005 universal testing machine (San Cugat del Vallés, Spain) provided with mechanical extensometer. A pulling rate of 100 mm/min was used. Five replicates were measured and averaged.

The contact angles on the PU film surfaces were measured at 21 °C by using bidistilled and deionized water in an ILMS goniometer (GBX Instruments, Bourg de Pèage, France). At least five water droplets of 4 µL were placed on different locations of each PU film surface, and the water contact angles were measured 30 s after water drop deposition.

The dispersion of the nanosilica in the PUs was determined by transmission electron microscopy (TEM) in a Jeol JEM-1400 Plus microscope (Jeol, Tokyo, Japan) by using an acceleration voltage of 120 kV.

#### 2.3.4. Adhesion Measurements

The adhesion of the PUDs was determined by T-peel tests of plasticized polyvinyl chloride (PVC)/PUD/plasticized PVC joints. The plasticized PVC test samples, with dimensions of 30 mm × 150 mm × 4 mm, were methyl ethyl ketone wiped for plasticizer removal. An amount of 3 mL PUD was applied by brush to each PVC strip and, after water evaporation at 25 °C for 1 h, the adhesive film was melted at 85 °C for 10 s under infrared radiation. The PVC strips were immediately placed in contact and a pressure of 0.4 MPa was applied for 10 s. The T-peel strength (Figure 2) was measured 15 min (immediate adhesion) and 72 h (final adhesion) after joint formation in a Zwick/Roell Z005 universal testing machine (San Cugat del Vallés, Spain). A crosshead speed of 100 mm/min was used. Five replicates were measured and averaged. The loci of failure of the joints were assessed by visual inspection.

## 3. Results and Discussion

The X-ray diffractogram of the nanosilica powder (Appendix A) shows one broad peak centered at 2θ value of 22 degrees. This is an indication of the amorphous structure of the nanosilica.

Even the nominal particle size of the nanosilica particles in the aqueous dispersion is 12 nm, the spherical nanosilica particles appear agglomerated in clusters of about 150 nm (Figure 3). The agglomeration of the nanosilica particles can be ascribed to the functionalization with acrylic moieties rather than to the interactions between the silanol groups on the surface of the nanosilica particles [38]. In fact, the chemical composition of the nanosilica powder obtained by IR spectroscopy (Figure 4) consists of typical bands of silica (Si-O bending at 798 cm^−1^, Si-O-Si stretching at 1106 and 962 cm^−1^) and another band at 1621 cm^−1^ due to C=C stretching [39]. Furthermore, the chemical composition of the nanosilica powder surface obtained from XPS experiments consists of 56.3 at.% oxygen, 40.2 at.% silicon, 2.5 at.% carbon, 0.9 at.% sodium and traces of chlorine (Appendix A). Therefore, the nanosilica powder surface is functionalized with carbon- and oxygen-containing species. In addition, according to the curve fitting of the C1s high-resolution XPS spectrum (Appendix A), these species are acrylic moieties because of the presence of 4 at.% C-O species at a binding energy of 287.0 eV and 16 at.% –O=C-OH species at a binding energy of 289.0 eV [40].

In order to allow an adequate dispersion into the PUD, the nanosilica particles in the aqueous dispersion should be separated and the most literature [30,31,32,41] have proposed severe mechanical stirring. However, in this study, a mild mechanical stirring of the aqueous nanosilica dispersion and the PUD was used, combined with the addition of small amounts (0.1–0.5 wt.%) of nanosilica. As a proof of concept, one additional PUD containing higher amounts of nanosilica (3 wt.%) was prepared.

### 3.1. Characterization of the Waterborne Polyurethane Dispersions (PUDs) without and with Different Amounts of Nanosilica

The dispersion of the nanosilica particles in the waterborne polyurethane was assessed by TEM micrographs (Figure 5). The phase separation in the PUD without nanosilica can be noticed in the TEM micrographs, in which the hard (dark zones) and soft (light zones) domains are distinguished [42]. The addition of nanosilica changes the degree of phase separation in the waterborne polyurethane to a greater extent by increasing the amount. The addition of only 0.1 wt.% nanosilica produces a good dispersion of the particles in the waterborne polyurethane matrix—the most particles are isolated and a few bundles of 3–4 particles are distinguished (Figure 5). Similarly, the addition of 0.5 wt.% nanosilica also allows a good dispersion of the nanosilica particles, but most of the nanosilica particles form bundles of 9–12 particles. However, the addition of 3 wt.% nanosilica produces agglomerates of nanosilica particles of about 150 nm in length and, in some minor zones, smaller clusters of nanosilica particles can be distinguished (Figure 5). Therefore, the physical mixing at a low stirring rate when 0.1–0.5 wt.% nanosilica is added seems sufficient for breaking the nanosilica agglomerates in the aqueous dispersion and a change in the degree of phase separation in the waterborne polyurethane is produced. However, the nanosilica bundles in the aqueous dispersion containing 3 wt.% nanosilica are not disagglomerated completely when a low stirring rate is applied, and separated domains of nanosilica particles and polyurethane can be distinguished.

Some properties of the waterborne polyurethane dispersions (PUDs) with and without different amounts of nanosilica are shown in Table 1. No sedimentation of the nanosilica particles in the PUDs was noticed over time. The solid content of the dispersions was 49–50 wt.% and decreased slightly by adding nanosilica because of the lower content of nanosilica in the aqueous dispersion (28.9 wt.%) than in the PUD (50 wt.%). The pH values of the PUDs are basic and increase gradually by increasing the amount of nanosilica because the pH of the nanosilica dispersion (9.2) is more basic than the one of the PUD without nanosilica (7.9). On the other hand, the surface tension of the PUDs increases by adding nanosilica and by increasing its amount because the surface tension of the nanosilica dispersion (63 mN/m) is higher than the one of the PUD without nanosilica (49 mN/m). Therefore, the higher the nanosilica content in the PUD, the higher the pH and the surface tension.

The magnitude of the zeta (Z) potential indicates the degree of electrostatic repulsion between adjacent, similarly charged particles in a dispersion. The Z-potential values of the PUDs are negative because the waterborne polyurethane is anionic and the nanosilica particles in the aqueous dispersion are negatively charged. The Z-potential values of the PUDs become more negative by adding nanosilica because of the more negative Z-potential value of the nanosilica dispersion (−87 mV) with respect to the one of the PUD without nanosilica (−57 mV). The more negative Z-potential value means a higher concentration of negative charges and higher resistance to the aggregation of the particles. On the other hand, the Z-potential values of the PUDs are similar irrespective of their nanosilica content (−63 to −65 mV). This indicates that the degree of nanosilica agglomeration does not affect their net negative charge.

Figure 6 shows the particle size distributions of the PUDs. The PUD without nanosilica shows a relatively narrow particle-size distribution, 67% made by particles of 217 nm, 22% particles of 25 nm and 11% particles of 297 nm. The addition of 0.1 wt.% nanosilica noticeably changes the particle-size distribution (Figure 6) because a decrease in the percentage of particles of 212 nm and an increase in the one of 298 nm particles is noticed (Appendix A). Considering that most of the nanosilica particles in the aqueous dispersion have mean particle sizes of 140 nm and 768 nm (Appendix A), the addition of 0.1 wt.% nanosilica to the PUD by using a mild physical mixing causes a good dispersion of the nanosilica particles (Figure 5). However, the addition of 0.5 and 3 wt.% nanosilica causes a moderate broadening of the particle size distribution of the PUD to lower and higher particle sizes, more noticeably by adding 3 wt.% nanosilica (Figure 6), the broadening is likely due to the existence of nanosilica agglomerates (Figure 5). Thus, the addition of 0.5 and 3 wt.% nanosilica decreases the percentage of particles of 202–209 nm and increases the ones of 265–266 nm. Their percentages are almost 50% (Appendix A). Thus, depending on the amount of nanosilica, the particle size distributions of the PUDs change differently.

Figure 7 shows the variation of the Brookfield viscosity of the PUDs as a function of the shear rate. Because the Brookfield viscosity of the nanosilica dispersion (11 mPa.s) is significantly lower than that of the PUD (140 nm), the addition of the nanosilica dispersion decreases the Brookfield viscosities of the PUDs. Whereas the Brookfield viscosities of the PUD without and with 0.1 wt.% nanosilica are similar, a decrease is produced by increasing the amount of nanosilica (Table 2). On the other hand, whereas the aqueous nanosilica dispersion shows a Newtonian rheological behavior, the PUDs show shear thinning, i.e., the viscosity decreases by increasing the shear rate (Figure 7). The extent of shear thinning can be quantified by the ratio of the Brookfield viscosities at 1 and 20 s^−1^ (pseudoplastic index). The pseudoplastic index and the extent of shear thinning of the PUDs decrease by adding nanosilica, more markedly by adding 3 wt.% nanosilica. Because the shear thinning in the waterborne polyurethane dispersion is caused by the reversible destruction of the interactions between the polyurethane particles, less shear thinning indicates the intercalation of the nanosilica particles between the polyurethane particles reducing their interactions, more efficiently by increasing the amount of nanosilica. Interestingly, the decrease in the viscosity of PUD+SiH(3) is not related to the extent of agglomeration/dispersion of the nanosilica particles.

### 3.2. Characterization of the Solid Waterborne Polyurethanes (PUs) without and with Different Amounts of Nanosilica

Upon evaporation of the water in the PUDs, solid-waterborne-polyurethane–nanosilica (PU+nanosilica) materials were obtained, and their structural, viscoelastic and mechanical properties were assessed.

The ATR-IR spectrum of the PU without nanosilica (Appendix A, Figure 8) shows the N-H stretching at 3400 cm^−1^, C=O stretching of urethane at 1730 cm^−1^, C-N and N-H bending at 1531 cm^−1^ and COO bending of the urethane group at 736 cm^−1^; all these bands correspond to the hard segments. Furthermore, different bands of the soft segments (C-H stretching at 2950, 2920 and 2870 cm^−1^, C-H bending at 1464 cm^−1^, C-H bending in CH_2_CO group at 1420 cm^−1^ and C-O-C stretching at 1238 and 1170 cm^−1^) can be distinguished. The ATR-IR spectra of the PU+nanosilica materials show more intense bands at 1110 and 1140 cm^−1^ due to the nanosilica, to a greater extent by increasing its amount. Some changes in the intensities of the C-H and C-O-C stretching bands of the soft segments of the polyurethane in the ATR-IR spectra are noticed, in a different manner depending on the amount of nanosilica. Thus, the addition of 0.1 wt.% nanosilica increases the C-H stretching band at 2920 cm^−1^, whereas the addition of 0.5 wt.% or more nanosilica does not change the intensities of the C-H stretching bands (Figure 8). However, while the addition of 0.1 wt.% nanosilica does not change the intensities of the C-O-C stretching bands, the addition of higher amounts of nanosilica changes the intensities of the bands at 1238 and 1170 cm^−1^ of the soft segments, more noticeably in PU+SiH(3). These changes are produced by the intercalation of the nanosilica particles between the soft segments, which produces a change in the degree of phase separation [43]. The change in the degree of phase separation can be estimated as the ratio of the intensities of the C-O-C stretching band of the soft segments at 1170 cm^−1^ and the C=O band of the hard segments at 1730 cm^−1^. Whereas the ratio of the intensities is 1.06 for the PU without nanosilica, lower values (0.85–0.95) are obtained in the PUs with 0.1 and 0.5 wt.% nanosilica. This confirms the phase separation due to the intercalation of the nanosilica particles between the polyurethane chains. However, similar ratios of the intensities of the C-O-C stretching and the C=O bands are obtained in the PU without and with 3 wt.% nanosilica because of the existence of nanosilica agglomerates.

The physical structures of the PU+nanosilica materials were assessed by DSC. The DSC curves of the first heating run of all PUs (Appendix A) show the glass transition (T_g_) of the soft segments at (−48)–(−50) °C and the melting of the soft segments. Whereas the addition of nanosilica does not change the T_g_ value of the soft segments, there are changes in the melting peak. While the addition of 0.1 wt.% nanosilica does not change the melting temperature and enthalpy of the PU without nanosilica, the addition of 0.5 wt.% nanosilica produces two melting peaks at 46 and 52 °C and increases the melting enthalpy. These two melting peaks should correspond to two different structures of the soft segments: one without and another with intercalated nanosilica. In contrast, the addition of 3 wt.% nanosilica displaces the melting to higher temperatures and the melting enthalpy is similar (Table 3). Therefore, the addition of 0.5 wt.% or more nanosilica produces two different structures of the soft segments, one without and another with intercalated nanosilica.

The DSC curves of the cooling run of the PU and PU+nanosilica materials show the crystallization of the soft segments of the polyurethane (Figure 9). Whereas in the PU without nanosilica the crystallization peak of the soft segments appears at −7 °C with a crystallization enthalpy of 30 J/g, the crystallization peak displaces to higher temperature (0–2 °C) and has higher crystallization enthalpy (33–35 J/g) in all PU+nanosilica materials, irrespective of the amount of nanosilica (Appendix A). This confirms the intercalation of the nanosilica particles between the soft segments of the polyurethane, which causes the formation of new crystallites.

After allowing a slow reorganization of the polyurethane chains, a second DSC heating run of the PU and PU+nanosilica materials was carried out (Figure 10). The DSC curves of all PUs show the glass transitions of the soft (T_g1_) and hard (T_g2_) segments, and the melting of the soft segments. Additionally, the DSC curves of PU and PU+SiH(0.5) show a cold crystallization at −21 °C with a small enthalpy (1.5–1.6 J/g) (Table 4). During the cold crystallization, at the transition zone between the existing crystalline structures and the amorphous regions, new ordered structures (crystallites) grow. These newly crystallites can be differentiated from the pre-existing ones by their lower melting temperatures. Whereas the DSC curves of PU and PU+SiH(0.5) are similar, they differ in PU+SiH(0.1) and PU+SiH(3), in which the cold crystallization is absent. The addition of 3 wt.% nanosilica causes similar thermal events as those caused by the PU without nanosilica, except for the higher melting enthalpy of the soft segments. However, the addition of 0.1 wt.% nanosilica mildly reduces the T_g1_ value and inhibits the appearance of the cold crystallization of the polyurethane. The absence of cold crystallization in the DSC curves of PU+SiH(0.1) and PU+SiH(3) can be ascribed to the disruption of the interactions between the soft segments caused by the intercalation of the nanosilica particles; unexpectedly, the cold crystallization remains in the DSC curve of PU+SiH(0.5). On the other hand, the glass transition temperature of the hard segments is not affected by adding nanosilica (236–237 °C) because of the lack of interactions between them. In summary, the amount of nanosilica affects the segmented structure of the polyurethane differently.

The TGA curve of the PU without nanosilica (Figure 11) shows two main thermal decompositions starting at 250 °C and 340 °C due to the hard and soft domains, respectively. The addition of nanosilica increases the thermal stability of the waterborne polyurethanes, irrespective of their nanosilica content, due to the existence of nanosilica–polyurethane interactions, in agreement with previous studies [30,37]. In fact, the temperatures at which 5 (T_5%_) and 50 (T_50%_) mass loss is produced are higher in all PU+nanosilica materials than in PU (Table 5); similar higher T_5%_ values are obtained. However, some differences in the T_50%_ values are noticed because of the changes in the physical structure of the soft segments by adding different amounts of nanosilica.

The physical structures in the PU and PU+nanosilica materials can be also assessed in the derivative of the TGA curves (Figure 12). It has been have demonstrated recently [7,44] that the thermal decompositions of the urethane and urea hard domains in the polyurethanes appear at 240–320 °C, and the one of the soft domains appears at 320–390 °C. All PUs show different thermal decompositions at 52–55 °C (residual water), 241–300 °C (urethane and urea hard domains), 329–384 °C (soft domains) and 436–442 °C (by-products formed during the TGA experiments [45]) (Appendix A). The most important weight loss corresponds to the soft domains at 329–384 °C. The soft domains appear at 329 °C in the PU without nanosilica and the PU+nanosilica materials show higher temperatures of decomposition of the soft domains at a different temperature and with different weight loss depending on their nanosilica content. This is an indication of the intercalation of the nanosilica particles between the soft domains of the polyurethane. The addition of 0.1–0.5 wt.% nanosilica displaces the temperature of decomposition of the soft domains of the PU to 361–369 °C with 60–73% weight loss—the thermal decomposition of the soft domains in the PU without nanosilica appears at 329 °C with a weight loss of 66% (Appendix A). Furthermore, the addition of 0.1–0.5 wt.% nanosilica increases the temperature of decomposition of the hard domains from 241 °C to 293–300 °C because a change in the degree of phase separation is produced (Appendix A). The amount of soft domains with intercalated nanosilica is higher and the one of the soft domains without nanosilica is lower in PU+SiH(0.1) than in PU+SiH(0.5) because of the more efficient dispersion of the nanosilica in PU+SiH(0.1). On the other hand, the addition of 3 wt.% nanosilica displaces the main peak of the DTGA curve to a higher temperature (Figure 12) and produces two kinds of soft domains at 356 °C and 384 °C, i.e., the temperature of the soft domains without nanosilica is higher and the one of the soft domains with intercalated nanosilica is lower (Appendix A). This is an indication of the worse dispersion of the nanosilica particles in the polyurethane matrix.

The structural changes caused in the polyurethane by adding nanosilica should affect their viscoelastic properties. As such, they were assessed by temperature-sweep plate–plate rheological experiments. The variation of the storage (G′) modulus of the PU and PU+nanosilica materials as a function of the temperature shows a continuous decrease in G′ by increasing the temperature (Figure 13). The rheological curves of all PUs are somewhat similar. However, the G′ values of the PUs increase slightly by increasing the nanosilica content. All PUs show a cross-over of the storage (G′) and loss (G″) moduli (Appendix A). Above the cross-over temperature (T_cross-over_), the PUs are mainly elastic, and below T_cross-over_, they are mainly viscous. The values of T_cross-over_ decrease and the moduli at the cross-over (G_cross-over_) of the PUs increase by adding nanosilica, to a greater extent by increasing the nanosilica content (Table 6). The lower T_cross-over_ value and the higher G_cross-over_ indicates higher interactions between the polyurethane chains caused by the intercalation of the nanosilica particles.

Previous studies are controversial with respect to the improvement of the mechanical properties of the waterborne polyurethanes by adding nanosilica. Whereas some literature [30,46] has shown improved mechanical properties of the PUs by adding silicas, other literature has evidenced the opposite trend, which was ascribed to the sedimentation of the silica particles [34]. In this study, no sedimentation of the nanosilica particles was found, so improved mechanical properties can be expected.

The stress–strain curves of the PU and PU+nanosilica materials show a marked yield point followed by an ample elastic deformation region (Figure 14); at a strain of 550% (the highest reached in the equipment), none of the PUs break, so the maximum strength was measured at a strain of 550%. Whereas the addition of 0.1 wt.% nanosilica increases moderately the yield strain and the stress at 550% of the PU, the addition of 0.5 wt.% nanosilica increases all mechanical properties. The improved mechanical properties of the PUs containing 0.1–0.5 wt.% nanosilica can be ascribed to the good dispersion of the nanosilica. However, the addition of 3 wt.% nanosilica increases the Young modulus and decreases the stress at 550% because of the existence of nanosilica agglomerates in the polyurethane.

### 3.3. Adhesion Properties of the Waterborne Polyurethane Dispersions (PUDs) without and with Different Amounts of Nanosilica

Because a suitable wettability of the adhesive is needed for an adequate adhesion, the water contact angles on the PU and PU+nanosilica surfaces were measured. The water contact angle on the PU surface is 68 degrees (Table 7), which seems adequate for producing good wettability. The addition of nanosilica increases the water contact angle values, more noticeably by increasing its amount. This trend agrees with the degree of dispersion of the nanosilica particles in the polyurethane matrix (the surface tension of the nanosilica dispersion (60 mN/m) is significantly higher than that of the waterborne polyurethane (49 mN/m)). Because the wettability of the polyurethane containing 3 wt.% nanosilica surface is worse than that of the PU surface, the acrylic moieties of the nanosilica dispersion seem to migrate to the surface. As a consequence, the adhesion of the waterborne polyurethanes is not expected to increase when the nanosilica dispersion is added.

The influence of the addition of the nanosilica dispersion to the waterborne polyurethane on its adhesion properties is not clear in the existing literature. Both increased adhesion [29,31,33] and decreased adhesion to different substrates [21] of silica–waterborne polyurethanes have been shown, the increased adhesion was related to the existence of interactions between the nanosilica and the polyurethane, and to the extent of the nanosilica agglomeration.

The adhesion properties of the waterborne polyurethane dispersions without and with different amounts of nanosilica were assessed by T-peel tests of plasticized PVC/PUD/plasticized PVC joints after 15 min (immediate adhesion) and 72 h (final adhesion) of joint formation (Figure 15). A short time (15 min) after joint formation is not sufficient for complete removal of the water in the PUDs and they are not completely cross-linked. Therefore, the T-peel strength values are somewhat low (1.8–2.5 kN/m) and the loci of failure in all joints are cohesive failures of the adhesive (Figure 16). The amount of nanosilica in the waterborne polyurethane determines the immediate T-peel strength because the joints made with the PUD containing 0.1 wt.% and 0.5 wt.% show similar immediate T-peel strength compared to the PUD without nanosilica, but the adhesion is lower in the joint made with the PUD with 3 wt.% nanosilica. Therefore, the presence of nanosilica agglomerates in the PUD decreases the immediate T-peel strength.

When the water in the PUD is completely removed (72 h after joint formation), the full cross-linking is produced, and the T-peel strength of the plasticized PVC/PUD/plasticized PVC joints (final adhesion) increases with respect to the T-peel strength values obtained 15 min after joint formation (Figure 15). The final T-peel strength values of the joints made with the PUDs containing nanosilica are significantly lower than the one obtained with the PUD without nanosilica. The higher the nanosilica content, the lower the final adhesion. The better the nanosilica dispersion in the PUD+nanosilica, the higher the final T-peel strength value. On the other hand, whereas the loci of failure of the joints made with the PUD without and with 0.1 wt.% nanosilica are cohesive ruptures of the PVC substrate (Figure 16), the one in the joint made with PUD+SiH(0.5) is mixed (cohesive rupture of the PVC + Cohesion in a surface layer of PVC) (Figure 16), and the one in the joint made with PUD+SiH(3) is cohesion in a surface layer of PVC (Figure 16). These experimental results agree with the increase in the surface tension of the PUDs containing nanosilica and with the extent of dispersion of the nanosilica in the polyurethane. However, the decrease in the final adhesion is too important to be ascribed to those factors only, and the main reason seems to be the migration of acrylic moieties of the nanosilica particles to the interface. In fact, the more dispersed the nanosilica particles, the higher the concentration of acrylic moieties on the adhesive surface. The deleterious adhesion of the waterborne polyurethane dispersions containing nanosilica has been ascribed previously to the migration of surfactant/antiadherent moieties to the interface, a change in the loci of failure from cohesive rupture to surface cohesive failure of the substrate was evidenced [47]. Similarly, it has been shown [20] that the adhesion of the waterborne polyurethane dispersions containing 1–5 wt.% nanosilica made by physical mixing was lower than in the dispersion without nanosilica.

## 4. Conclusions

Waterborne polyurethane dispersions containing different small amounts of nanosilica have been successfully prepared by physical mixing using mild mechanical stirring. The PUD+nanosilica dispersions were stable and showed no sedimentation over time.

The addition of 0.1–0.5 wt.% nanosilica produced a good dispersion of the particles in the waterborne polyurethane matrix, and the most particles were isolated. However, the addition of 3 wt.% nanosilica produced agglomerates of nanosilica particles of about 150 nm length. Therefore, when small amounts of nanosilica were added, the physical mixing at a low stirring rate was sufficient for breaking the nanosilica agglomerates in the aqueous dispersion, and a change in the degree of phase separation in the waterborne polyurethane was produced. 

The higher the nanosilica content in the PUD, the higher the pH and the surface tension. Depending on the amount of nanosilica, the particle size distributions of the PUDs changed differently and the addition of only 0.1 wt.% nanosilica noticeably increased the percentage of the particles of 298 nm. However, the addition of 0.5 and 3 wt.% nanosilica caused a moderate broadening of the particle-size distribution of the PUD, more noticeably by adding 3 wt.% nanosilica. On the other hand, the Brookfield viscosity of the PUD decreased by increasing the amount of nanosilica above 0.5 wt.% and all PUDs showed shear thinning, the extent of shear thinning decreased by adding nanosilica and by increasing its amount.

The addition of 0.1–0.5 wt.% nanosilica increased the degree of phase separation due to the intercalation of the nanosilica particles among the polyurethane chains. As a consequence, the DSC curves showed two melting peaks at 46 °C and 52 °C, as well as an increase in the melting enthalpy, and the crystallization peak of the polyurethane was displaced to higher temperature and showed higher enthalpy. On the other hand, the addition of 0.1–0.5 wt.% nanosilica displaced the temperature of decomposition of the soft domains to a higher temperature and increased the temperature of decomposition of the hard domains due to the change in the degree of phase separation. However, the addition of 3 wt.% nanosilica produced two kinds of soft segments: one without nanosilica and another with intercalated nanosilica. They are caused by the worse dispersion of the nanosilica particles in the polyurethane. Furthermore, the values of the temperature at the cross-over of the storage and loss moduli decreased, and the moduli at the cross-over increased to a greater extent by increasing the nanosilica content.

The addition of nanosilica increased the thermal stability of the polyurethanes, irrespective of their nanosilica content, and the mechanical properties were improved by adding 0.1–0–5 wt.% nanosilica.

The addition of nanosilica decreased the wettability of the waterborne polyurethanes to a greater extent by increasing the amount. The amount of nanosilica in the waterborne polyurethane determined the immediate T-peel strength because the joints made with the PUD containing 0.1 and 0.5 wt.% showed similar immediate T-peel strength compared to the PUD without nanosilica, but the adhesion was lower in the joint made with the PUD with 3 wt.% nanosilica. The final T-peel strength values of the joints made with the PUDs containing nanosilica were significantly lower than the obtained with the PUD without nanosilica. The higher the nanosilica content, the lower the final adhesion. The better the nanosilica dispersion in the PU+nanosilica, the higher the final T-peel strength value. The decreased final adhesion was ascribed to the migration of acrylic moieties of the nanosilica particles to the interface and the decreased wettability of the PUDs containing nanosilica.

## Figures and Tables

**Figure 1 polymers-14-05136-f001:**
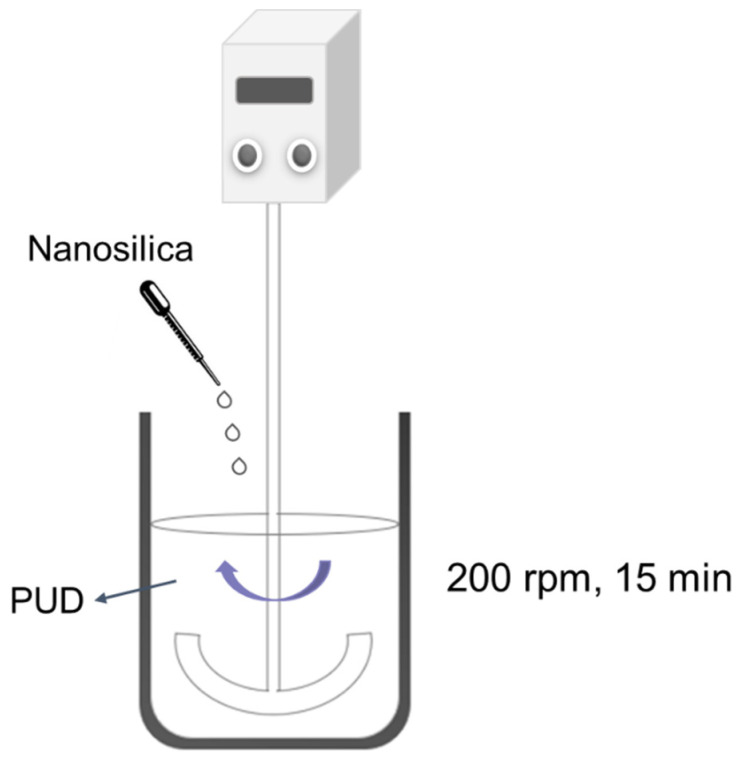
Physical mixing of the waterborne polyurethane dispersion (PUD) and the aqueous nanosilica dispersion.

**Figure 2 polymers-14-05136-f002:**
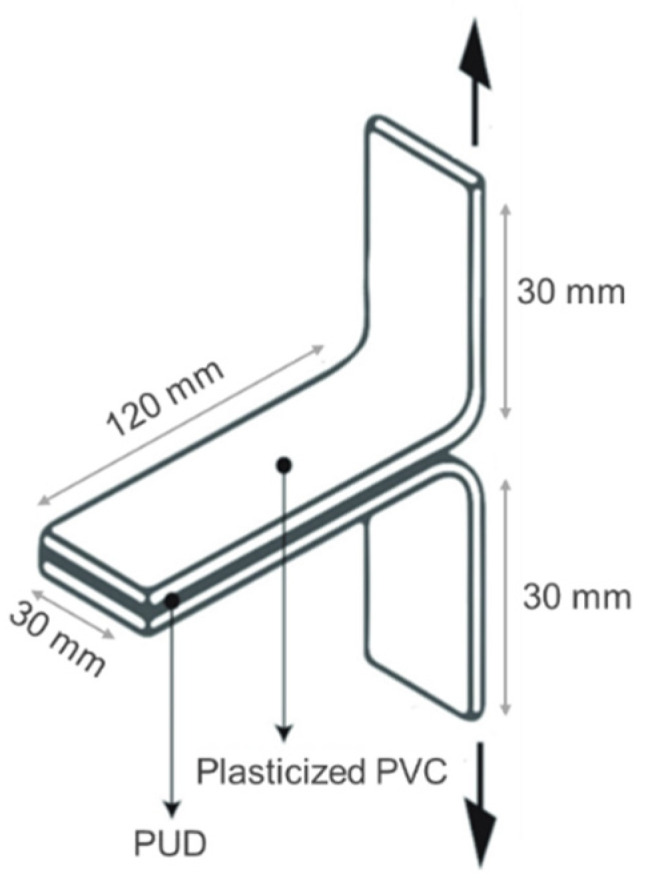
Adhesion test used in this study.

**Figure 3 polymers-14-05136-f003:**
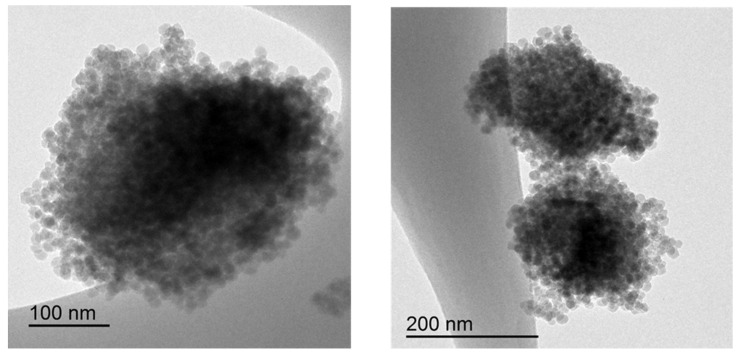
TEM micrographs of the nanosilica particles. The carbon grid can be distinguished in the TEM micrographs.

**Figure 4 polymers-14-05136-f004:**
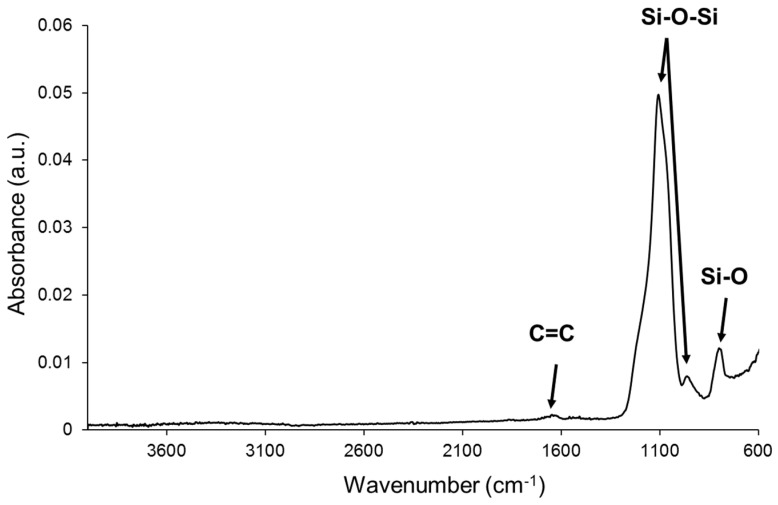
ATR-IR spectrum of the nanosilica powder.

**Figure 5 polymers-14-05136-f005:**
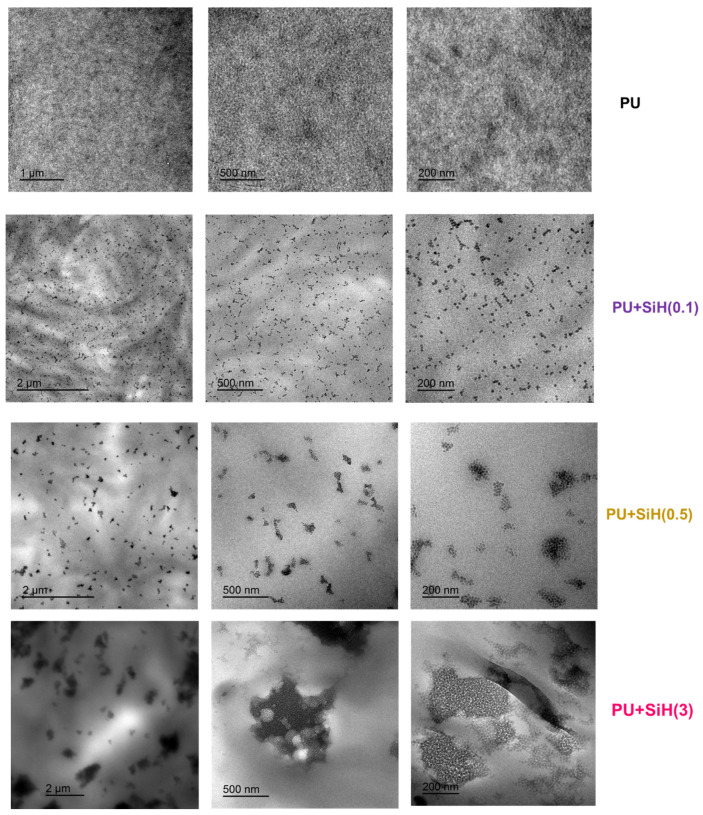
TEM micrographs at different magnifications of the waterborne polyurethane dispersions without and with different amounts of nanosilica.

**Figure 6 polymers-14-05136-f006:**
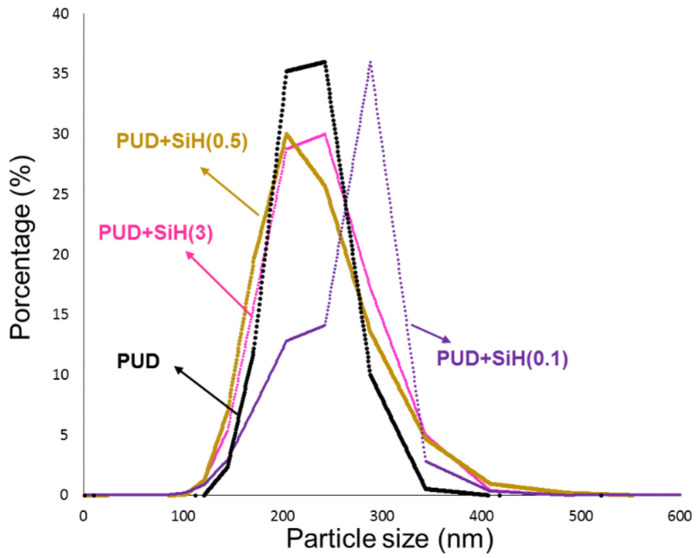
Particle size distributions of the waterborne polyurethane dispersions without and with different amounts of nanosilica.

**Figure 7 polymers-14-05136-f007:**
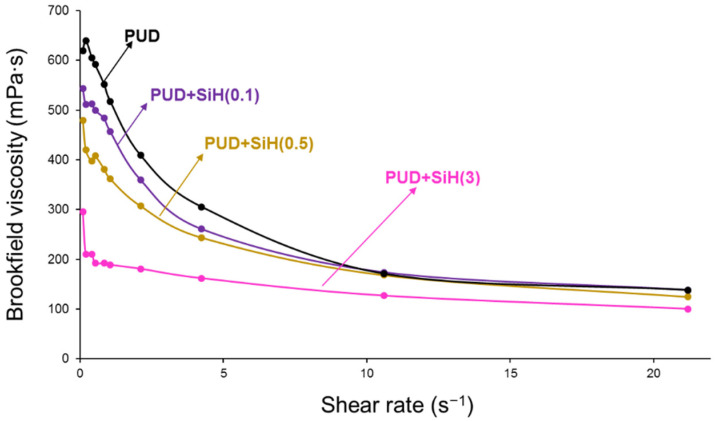
Variation of Brookfield viscosity of the waterborne polyurethane dispersions without and with different amounts of nanosilica as a function of the shear rate.

**Figure 8 polymers-14-05136-f008:**
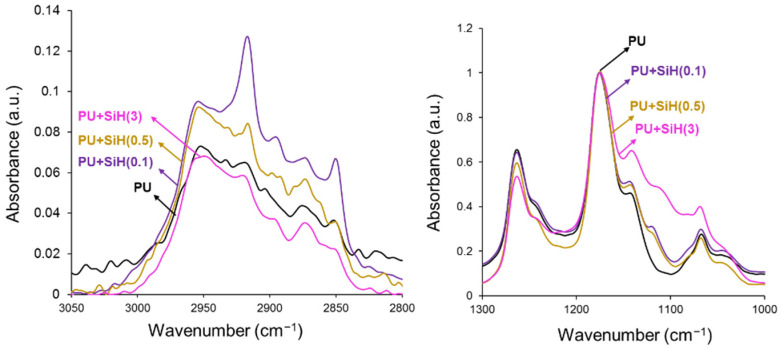
Different regions of the ATR-IR spectra of the PU and PU+nanosilica materials.

**Figure 9 polymers-14-05136-f009:**
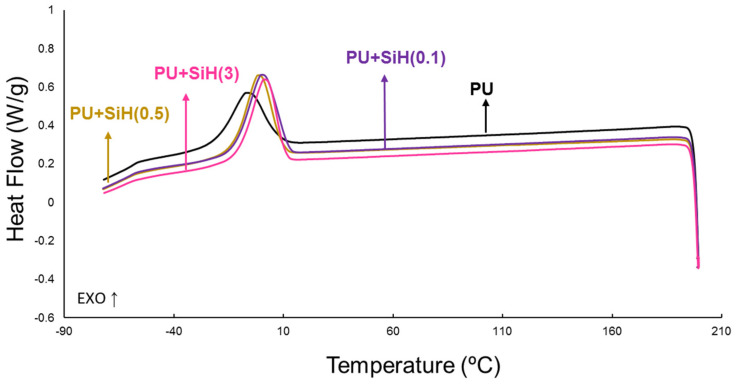
DSC curves of the PU and PU+nanosilica materials. Cooling run.

**Figure 10 polymers-14-05136-f010:**
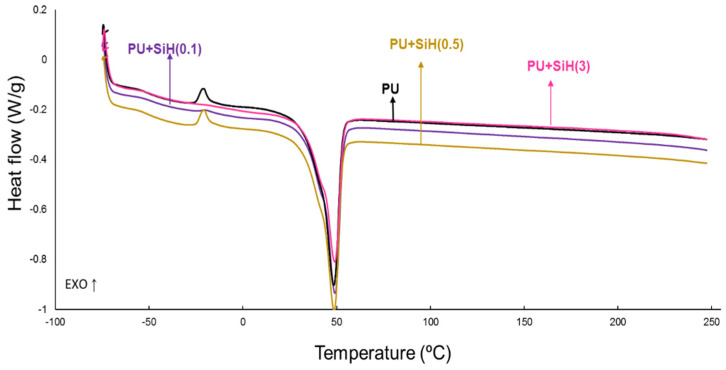
DSC curves of the PU and PU+nanosilica materials. Second heating run.

**Figure 11 polymers-14-05136-f011:**
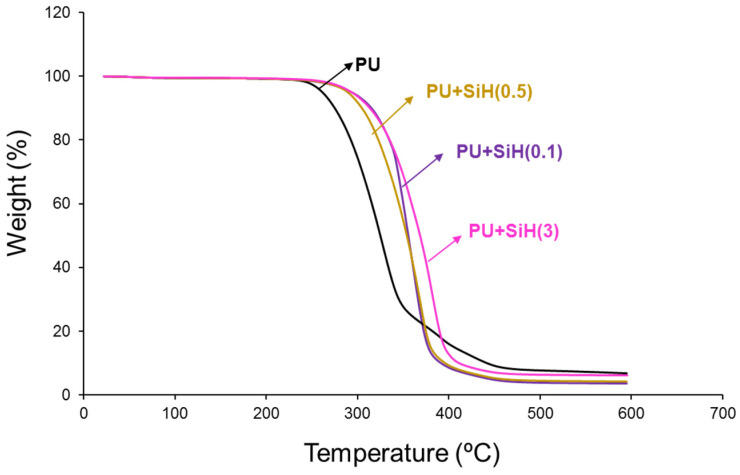
TGA curves of the PU and PU+nanosilica materials.

**Figure 12 polymers-14-05136-f012:**
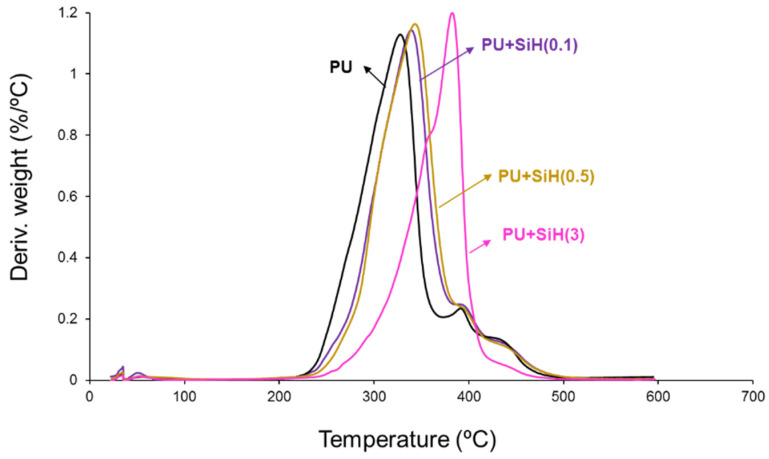
Derivative of the TGA curves of the PU and PU+nanosilica materials.

**Figure 13 polymers-14-05136-f013:**
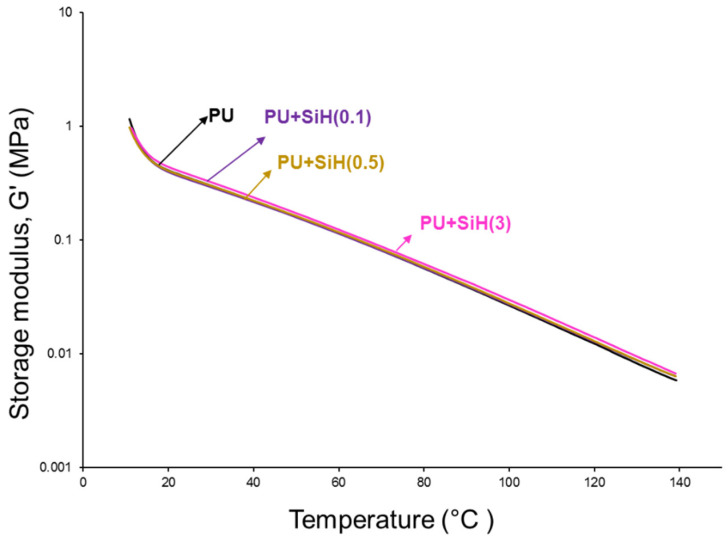
Variation of the storage modulus (G′) as a function of the temperature for PU and PU+nanosilica materials.

**Figure 14 polymers-14-05136-f014:**
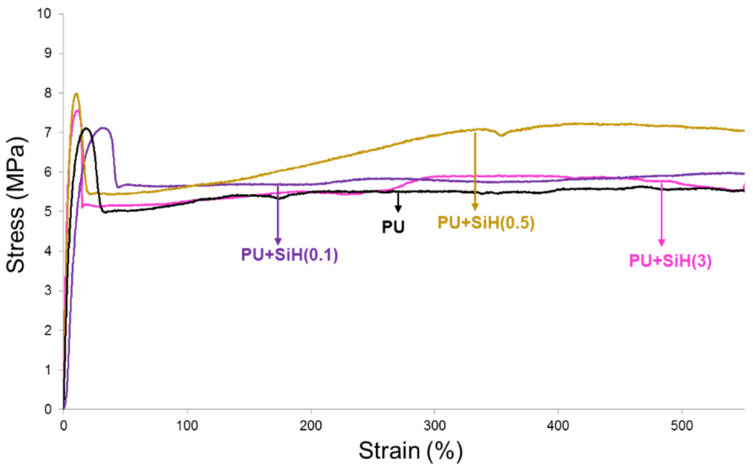
Stress–strain plots of the PU and PU+nanosilica materials.

**Figure 15 polymers-14-05136-f015:**
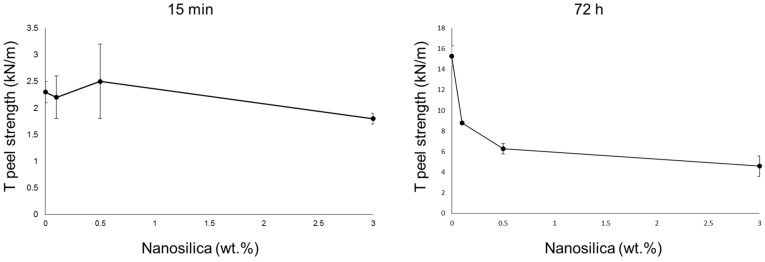
Variation of the T-peel strength values of plasticized PVC/waterborne polyurethane/plasticized PVC joints at different times after joints formation as a function of the nanosilica content in the waterborne polyurethane.

**Figure 16 polymers-14-05136-f016:**
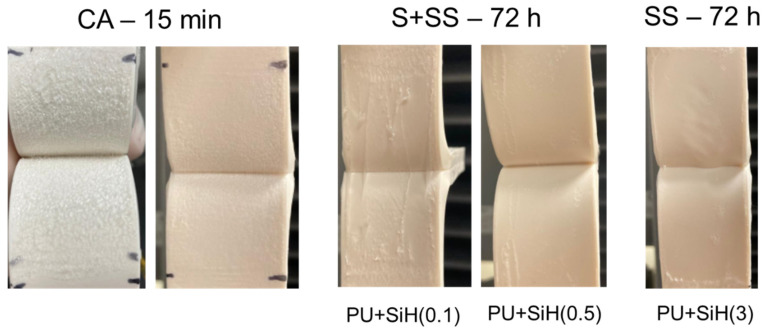
Loci of failure of plasticized PVC/polyurethane dispersion/plasticized PVC joints. Cohesive rupture of adhesive (CA); Substrate cohesive failure (S); Cohesion in a surface layer of the substrate (SS).

**Table 1 polymers-14-05136-t001:** Some properties of the waterborne polyurethane dispersions without and with different amounts of nanosilica.

Dispersion	Solids Content (wt.%)	pH	Z Potential (mV)	Surface Tension (mN/m)
PUD	49.7 ± 1.0	7.9	−57	49
PUD+SiH(0.1)	49.8 ± 0.3	8.0	−63	50
PUD+SiH(0.5)	49.4 ± 1.0	8.1	−65	51
PUD+SiH(3)	49.2 ± 0.9	8.3	−64	53

**Table 2 polymers-14-05136-t002:** Brookfield viscosities and pseudoplastic indexes of the waterborne polyurethane dispersions without and with different amounts of nanosilica.

Dispersion	Viscosity at 20 s^−1^ (mPa∙s)	Pseudoplastic Index
PUD	140	4.6
PUD+SiH(0.1)	140	3.7
PUD+SiH(0.5)	128	3.4
PUD+SiH(3)	103	2.1

**Table 3 polymers-14-05136-t003:** Some parameters obtained from the DSC curves of the PU and PU+nanosilica materials. First heating run.

Property	PU	PU+SiH(0.1)	PU+SiH(0.5)	PU+SiH(3)
T_g_ (°C)	−49	−50	−50	−48
T_m_ (°C)	48–50	48–51	46, 52	53
∆H_m_ (J/g)	41	41	44	45

**Table 4 polymers-14-05136-t004:** Some parameters obtained from the DSC curves of the PU and PU+nanosilica materials. Second heating run.

Property	PU	PU+SiH(0.1)	PU+SiH(0.5)	PU+SiH(3)
T_g1_ (°C)	−50	−47	−50	−49
T_g2_ (°C)	237	236	237	236
T_c_ (°C)	−21	-	−21	-
∆H_c_ (J/g)	1.5	-	1.6	-
T_m_ (°C)	48	49	48	48
∆H_m_ (J/g)	40	38	40	43

**Table 5 polymers-14-05136-t005:** Temperatures at which 5 (T_5%_) and 50 (T_50%_) mass loss is produced in the PU and PU+nanosilica materials. TGA experiments.

Polyurethane	T_5%_ (°C)	T_50%_ (°C)
PU	262	325
PU+SiH(0.1)	294	355
PU+SiH(0.5)	295	356
PU+SiH(3)	295	368

**Table 6 polymers-14-05136-t006:** Moduli and temperatures at the cross-over of the storage (G′) and loss (G″) moduli in the PU and PU+nanosilica materials. Plate–plate rheology experiments.

Polyurethane	G_cross-over_ (MPa)	T_cross-over_ (°C)
PU	5.7 × 10^−2^	84
PU+SiH(0.1)	6.0 × 10^−2^	78
PU+SiH(0.5)	6.2 × 10^−2^	80
PU+SiH(3)	6.9 × 10^−2^	77

**Table 7 polymers-14-05136-t007:** Some mechanical properties and water contact angle values of the PU+nanosilica materials. Stress–strain experiments.

Property	PU	PU+SiH(0.1)	PU+SiH(0.5)	PU+SiH(3)
Young modulus (MPa)	0.9 ± 0.0	0.9 ± 0.1	1.2 ± 0.2	1.5 ± 0.3
Yield stress (MPa)	7.0 ± 0.2	7.0 ± 0.4	7.9 ± 0.5	7.5 ± 0.4
Stress at 550% (MPa)	5.5 ± 0.3	5.9 ± 0.5	7.1 ± 0.2	5.6 ± 0.1
Water contact angle (degrees)	68	68	70	73

## Data Availability

Not applicable.

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
