# Peer review of "Efficient Physical Mixing of Small Amounts of Nanosilica Dispersion and Waterborne Polyurethane by Using Mild Stirring Conditions"

_polymers, 2022, doi:10.3390/polym14235136_

Round 1
Reviewer 1 Report
Attached you can find my comments.

Reviewer 2 Report
The author has worked and investigated the effect of the physical mixing of small amounts of nano silica dispersion and waterborne polyurethane by using mild stirring conditions. The work is interesting and can be accepted after a few modifications:
Abstract: The abstract needs to be more quantitative, depicting the results obtained in the work.
Introduction section: The section is well written. However, it lacks recent literature and a paragraph on nano-silica, its unique properties, and the reason for mixing with waterborne polyurethane. Some of the literature can be found as:
1) Khanna S, Marathey P, Chaliyawala H, Rajaram N, Roy D, Banerjee R, Mukhopadhyay I. Fabrication of long-ranged close-packed monolayer of silica nanospheres by spin coating. Colloids and Surfaces A: Physicochemical and Engineering Aspects. 2018 Sep 20;553:520-7.
2) Peruzzo PJ, Anbinder PS, Pardini FM, Pardini OR, Plivelic TS, Amalvy JI. On the strategies for incorporating nanosilica aqueous dispersion in the synthesis of waterborne polyurethane/silica nanocomposites: Effects on morphology and properties. Materials Today Communications. 2016 Mar 1;6:81-91.
3) Zhang S, Chen Z, Guo M, Bai H, Liu X. Synthesis and characterization of waterborne UV-curable polyurethane modified with side-chain triethoxysilane and colloidal silica. Colloids and Surfaces A: Physicochemical and Engineering Aspects. 2015 Mar 5;468:1-9.
Result and Disscusion:
1) Author needs to add X-ray, FESEM, EDX mapping and Raman spectroscopy measurements for the nano-silica.
2) What are the Z potential values of nano-silica? What does negative signify?
3) In comparison to the TGA graph, the derivative of the TGA curve shows a clear red shift for PU+SiH(3). Why is there a significant change?
4) Did the author perform DSC cyclability test for the stability of the PU+SiH composite? How many cycles and degradation?
4) Conclusion: The conclusion is well written.
A few grammatical errors need to be addressed in the revised version of the manuscript.
Round 2
Reviewer 1 Report
The authors significantly improved their manuscript
Reviewer 2 Report
The authors have addressed all the comments and thus the revised manuscript can be acceptable in the present form.